# Research Progress of Conjugated Nanomedicine for Cancer Treatment

**DOI:** 10.3390/pharmaceutics14071522

**Published:** 2022-07-21

**Authors:** Bin Zhao, Sa Chen, Ye Hong, Liangliang Jia, Ying Zhou, Xinyu He, Ying Wang, Zhongmin Tian, Zhe Yang, Di Gao

**Affiliations:** 1The Key Laboratory of Biomedical Information Engineering of Ministry of Education, School of Life Science and Technology, Xi’an Jiaotong University, Xi’an 710049, China; billness@stu.xjtu.edu.cn (B.Z.); chensakmc@stu.xjtu.edu.cn (S.C.); jll.15229619335@stu.xjtu.edu.cn (L.J.); zhouying123@stu.xjtu.edu.cn (Y.Z.); hexinyu@stu.xjtu.edu.cn (X.H.); wy1127@stu.xjtu.edu.cn (Y.W.); zmtian@mail.xjtu.edu.cn (Z.T.); 2Department of Epidemiology, Shaanxi Provincial Cancer Hospital, Xi’an 710061, China; 3Shaanxi Provincial Centre for Disease Control and Prevention, Xi’an 710054, China; 4Center of Digestive Endoscopy, Shaanxi Provincial Cancer Hospital, Xi’an 710061, China; sxchhy@163.com; 5Research Institute of Xi’an Jiaotong University, Hangzhou 311200, China

**Keywords:** drug-conjugates, conjugated nanomedicine, cancer therapy, prodrug, synergistic chemotherapy

## Abstract

The conventional cancer therapeutic modalities include surgery, chemotherapy and radiotherapy. Although immunotherapy and targeted therapy are also widely used in cancer treatment, chemotherapy remains the cornerstone of tumor treatment. With the rapid development of nanotechnology, nanomedicine is believed to be an emerging field to further improve the efficacy of chemotherapy. Until now, there are more than 17 kinds of nanomedicine for cancer therapy approved globally. Thereinto, conjugated nanomedicine, as an important type of nanomedicine, can not only possess the targeted delivery of chemotherapeutics with great precision but also achieve controlled drug release to avoid adverse effects. Meanwhile, conjugated nanomedicine provides the platform for combining several different therapeutic approaches (chemotherapy, photothermal therapy, photodynamic therapy, thermodynamic therapy, immunotherapy, etc.) with the purpose of achieving synergistic effects during cancer treatment. Therefore, this review focuses on conjugated nanomedicine and its various applications in synergistic chemotherapy. Additionally, the further perspectives and challenges of the conjugated nanomedicine are also addressed, which clarifies the design direction of a new generation of conjugated nanomedicine and facilitates the translation of them from the bench to the bedside.

## 1. Introduction

Cancer is the leading cause of death worldwide and a critical barrier to increasing life expectancy. According to the estimation of the World Health Organization (WHO), nearly 19.3 million new cases and 10 million deaths are closely related to cancer globally in 2020 [1]. Although new types of systemic cancer therapy (e.g., immunotherapy and targeted therapy) have been developed over the past few years, it is undeniable that chemotherapy still occupies a crucial position [1,2]. However, conventional chemotherapy suffers from intrinsic limitations, such as serious side effects, intrinsic and acquired multidrug resistance (MDR) and poor targeting capacity, causing more than 90% of new drugs to be abandoned before or during clinical trials [3]. Hence, the technologies that effectively improve the pharmacokinetics and the tumor accumulation of chemotherapeutic drugs are urgently needed [3,4]. Thereinto, employing carriers to deliver anti-cancer drugs to tumor tissues is a promising and innovative approach to increasing the efficacy of drugs while avoiding adverse effects [5,6,7].

Nanotechnology has been regarded as the next revolution to influence various industrial fields, including biomedicine. Especially, nanomedicine for cancer treatment has achieved tremendous progress in the last few decades. Due to unique physical and chemical properties, nanomedicine can enhance the pharmacokinetics of drugs via improving their stability, solubility, circulating half-life, and targeting capacity, and overcome drug resistance, finally resulting in enhanced chemotherapeutic efficacy. With the gradual maturity of nanomedicine research, nanomedicine has gradually entered clinical tumor treatment. Since the first cancer nanomedicine, Doxil, was approved by the Food and Drug Administration (FDA) in 1995, there have been at least 17 kinds of cancer nanomedicine applied in the clinic, including Abraxane (albumin-bound paclitaxel), Oncaspar (PEGylated L-asparaginase), Genexol-PM (mPEG-PDLLA micellar paclitaxel), NanoTherm^®^ AS1, etc. (Table 1). In addition to these commercial nanomedicines, a vast number of chemotherapeutic nanomedicines are undergoing clinical trials and experimental studies currently, encompassing many types of nanomedicine, including liposomes, polymer-based nanoparticles, albumin nanoparticles as well as inorganic nanoparticles [8,9].

Conjugated nanomedicine, formulated from drug conjugates that connect drugs via chemical linkers, has attracted more attention due to its unique potential in cancer treatment, which are listed as follows: 1. It increases the drug-carrying capacity of chemotherapeutic drugs. Both hydrophilic and hydrophobic substances nanomedicine can selectively deliver chemotherapeutics to tumor cells, even intracellular organelles via the passive and active targeting capacity, especially ones containing targeting ligands. 2. Several stimuli-responsive chemical bonds can be exploited as linkers to form drug conjugates, with the purpose of selective and controlled drug release. 3. Some medical imaging agents can also be conjugated with therapeutic agents for a diagnostic function. 4. Since fewer excipients without therapeutic effects are required, the biohazard caused by the degradation of the excipients can be effectively minimized. 5. The conjugated nanomedicine is also easily mass-produced due to the simple structure and easy preparation of drug conjugates. Based on these advantages, conjugated nanomedicine is considered a hopeful strategy to ameliorate cancer treatment outcomes in the future. Indeed, in recent years, conjugated nanomedicine has obtained more and more attention and the number of publications per year with the keyword “conjugate nanomedicine” maintains a high level (Figure 1, from the web of science database). Therefore, this review provides an overview of conjugated nanomedicines, summarizing the types of drug conjugates and representative applications of these conjugated nanomedicines in synergistic chemotherapy. Furthermore, the further perspectives and challenges of the conjugated nanomedicine are also addressed, which clarifies the design direction of a new generation of conjugated nanomedicine and facilitates the translation of them from bench to bedside (Figure 2).

## 2. Categories of Drug Conjugates

Conjugated nanomedicine formulated by drug conjugates has proved to enhance the therapeutic effect of drugs through improving their transmission to the disease sites by the virtue of the ligand-mediated active targeting process and enhanced permeability and retention (EPR) effect-based passive targeting process. Besides, the physicochemical properties of nanomedicine, such as size, shape, and surface chemical properties also determine the accumulation and deep penetration of nanoparticles into tumor tissues [15]. Besides, the controllable release of drugs under certain internal/external stimuli is also crucial for the desired therapeutic outcomes. To develop conjugated nanomedicine with properties as required, the design and preparation of promising drug conjugates is the first step toward success. The following sections will discuss the categories of drug conjugates in detail along with several representative examples.

### 2.1. Polymer-Drug Conjugates

Polymer-drug conjugates, also known as polymer prodrugs, are usually composed of polymer skeletons, chemical linkers (cleavable or uncleavable) and therapeutic drugs [16]. The first study of polymer-drug conjugates can be traced back to 1955 when Von Horst Jatzkewitz found that mescaline, a kind of psychedelic alkaloid, presented biological activity in mice even being coupled with poly (vinylpyrrolidone), accompanied by an extended residence time of mescaline in vivo [17]. In the 1970s, Ringsdorff, Kopecek and Duncan pioneered the development of therapeutic strategies based on polymer-drug conjugates [18,19]. Thereinto, Ringsdorff put forward the concept of a pharmacologically active polymer carrier, which can improve drug solubility and control drug release in a targeted way. About twenty years later, a new kind of polymer–protein conjugate named Adagen^®^ (pegademase bovine) was first approved by FDA for enzyme replacement therapy against severe combined immunodeficiency disease (SCID) associated with a deficiency of adenosine deaminase [18,19]. Since then, this drug delivery strategy received much more attention and a series of polymer–drug conjugates have been prepared for treating various illnesses. Furthermore, these conjugates could also be formulated into nanomedicines with types of polymer capsules [20,21], polymeric nanoparticles [22,23,24], dendrimers [25,26] and so on. 

As for the historical development of polymer–drug conjugates in the field of cancer treatment, Matsumura and Maeda reported the antitumor carcinogen SMANCS, a conjugate of partially half-butyl-esterified styrene-co-maleic acid polymer[butyl-SMA] and neocarzinostatin (NCS), which preferentially accumulated in tumor tissues after intravenous administration in the 1980s [27]. Besides, PK1 (FCE28068) is the first water-soluble polymer small molecular drug conjugate applied in clinical trials. It is composed of N-(2-hydroxypropyl) methacrylamide (HPMA) and doxorubicin (DOX) linked via a peptide-based lysosome-cleavable bond. Based on the results from clinical trials, PK1 displays a prolonged half-life of DOX and improved biosafety as compared with parent DOX treatment. It also displays the antineoplastic activity against breast cancer and non-small cell lung cancer (NSCLC) during Phase II clinical studies [28]. However, the tumor accumulation of PK1 is still insufficient, compromising the final therapeutic efficacy and inducing the termination of PK1 development [27]. Subsequently, Jameson and co-workers formed the conjugate Onzeald of tetrapod polyethylene glycol(PEG) and irinotecan (antineoplastic enzyme inhibitor) through lipid bonds. During a preclinical study, the half-life and tumoral concentration of irinotecan increased as compared with parent irinotecan, leading to an inhibited tumor growth and improved therapeutic index [29]. Learning from the above examples the majority of cytotoxic chemotherapeutic drugs present problems including poor aqueous solubility, limited tumor exposure and off-target toxicity. Once the drugs are linked to polymer carriers, their aqueous solubility and stability can be improved along with modified pharmacokinetics. Moreover, it also shows the potential for polymer–drug conjugates to precisely deliver drugs to tumor tissues via passive targeting and/or active targeting effects, and overcoming the undesired side effects of cytotoxic chemotherapeutics against healthy tissues [30]. In addition, these conjugated polymeric nanoparticles also displayed advantages, including overcoming MDR and reducing immunogenicity [31]. 

In the past decade, many polymer-drug conjugates have been commercially available and some others are undergoing clinical trials [30], on account of the rapid advancement of polymeric materials. Among them, PEG occupies the important position, which are represented by a series of clinically applied PEG-drug conjugates (e.g., Adynovate^®^, Oncaspar^®^, Plegridy^®^, etc.) [32,33]. Additionally, some synthetic biodegradable polymers bearing minimal long-term side effects and systemic toxicity, especially the ones approved by the FDA, have also been used to prepare the polymer-drug conjugates, such as poly(lactic-co-glycolic acid) (PLGA) [34], poly(lactic acid) (PLA) [35], poly(ε-caprolactone) (PCL) [36], poly(amino acid) (PAA) [37] and so on. For example, the allyl functionalized PLA is used as the precursor of the polymer backbone. A UV-induced mercaptan-alkene reaction was carried out to combine sulfobetaine (SB) with a PLA-based backbone to yield the carrier materials, followed by the encapsulation of paclitaxel (PTX). This drug delivery system based on polymer–drug coupling showed complete degradability, continuous drug release ability and significant anticancer effect [38]. PAA has also been widely used in the field of drug delivery systems due to its good biocompatibility, biodegradability and functionalization [39]. Because there are many active functional groups on the side chain of PAA, the drugs and bioactive molecules could be easily conjugated to PAAs, followed by preparation into nanoparticles for cancer treatment [40]. For example, Ma and co-workers conjugated dexamethasone (DEX, an anti-inflammatory agent) onto the side chain of mPEG-*b*-poly(_L_-lysine) (mPEG-*b*-PLL) via a redox and pH dual sensitive linker. The resultant polymer–drug conjugates were further self-assembled into micelles for the treatment of colorectal cancer. According to their results, cyclooxygenase-2 (COX-2) and α-smooth muscle actin (α-SMA) were dramatically decreased during therapy and the immunosuppressive microenvironment of the CT26 tumor was also relieved, resulting in the compromised tumor-promoting inflammation. It also provided a promising option for applying anti-inflammatory drugs for cancer treatment [41]. Natural polysaccharides, including chondroitin sulfate (CS), hyaluronic acid (HA), pullulan (PUL), and heparin (HEP) are also good candidates for conjugating drugs due to their outstanding virtues, such as biocompatibility, biodegradability, non-immunogenicity and toxicity, easy chemical modification, and low cost [42,43,44,45,46]. The antimalarial drug dihydroartemisinin (DHA) can inhibit cancer cell proliferation and induce apoptosis, while it has been associated with some limitations, such as poor aqueous solubility and rapid metabolism in the systemic circulation. Robin and co-workers successfully synthesized a HA–DHA conjugate via the conjugation of the carboxylic group of HA with the hydroxyl group of DHA. The enhanced cytotoxicity of nanoparticles against lung cancer (A549 cell line) was supported by the generation of reactive oxygen species (ROS), loss of mitochondrial membrane potential and exhibition of better cytotoxicity than native DHA [47]. 

### 2.2. Antibody-Drug Conjugates

Antibody–drug conjugates (ADC) are conjugates that couple specific monoclonal antibodies with cytotoxic drugs through specific linkages. They are targeted biological agents, which can selectively transport cytotoxic drugs to tumor cells with monoclonal antibodies as navigation. ADC generally includes the following three components: antibodies with high specificity and affinity to targets, connectors with high stability and small-molecule cytotoxic drugs with promising therapeutic efficacy [48]. ADC selectively delivers cytotoxic drugs to tumor cells by using the specific binding ability of monoclonal antibodies to cell surface target antigens. After ADC enters the body, it recognizes the specific antigen on the cell surface through autoantibody components. Then, the ADC antigen complex can be internalized by tumor cells via receptor-mediated endocytosis, followed by being degraded by intracellular enzymes and lysosomes. Cytotoxic agents are then released in the cytoplasm, finally causing cell death through an induction mechanism [49] (Figure 3). Moreover, chemical connectors in ADC with promising stimuli-responsiveness have been recognized as the prerequisites for minimizing the premature drug release in plasma and promoting the controllable release of payloads to cancer cells, which kindles the enthusiasm of researchers for constituting sensitive ADC, especially in the field of tumor precision therapy for decades.

At present, FDA has approved a series of ADCs for clinical cancer treatment, and many other ADCs are undergoing clinical trials [50]. As for traditional chemotherapy, anticancer drugs mainly execute rapidly dividing cells, but undesirable toxicity to normal cells is inevitable, resulting in high side effects. To solve this problem, the first generation of ADCs was developed by connecting cytotoxic drugs with monoclonal antibodies (mAb). These ADCs can target cancer cells and then selectively destroy them with lower side effects on healthy tissues. Although the role of monoclonal antibodies in cancer treatment was not fully understood at that time, inspired by the fact that many antibodies can preferentially bind to tumor cells, people connected anticancer drugs, such as melphalan, vinblastine, methotrexate, and DOX to monoclonal antibodies to form the first generation of ADCs. In 2000, the first ADC, gemtuzumab ozogamicin (GO), was approved by FDA, mainly for the treatment of patients with acute myeloid leukemia (AML). GO is a conjugate of an anti-CD33 mAb and calicheamicin, which are connected via the acid-cleavable hydrazone linkers. However, the results from the required post-approval study demonstrated the inadequate improvement of the survival rate and more severe toxicity of GO-based synergistic therapy over chemotherapy alone, forcing Pfizer to withdraw GO from the market in 2010 [51]. Some other studies also pointed out the drawbacks of first-generation ADCs, including poor potency of the cytotoxic drug, low localization of monoclonal antibodies and poor stability of linkages [50,51,52,53,54,55] Moreover, an undesirable immune response could be induced during ADCs-mediated therapy, which is caused by antibody components, rather than cytotoxic drugs [56]. 

The second generation of ADCs were optimized by using more effective tubulin targeting agents (e.g., monomethyl auristatin E (MMAE), maidenlig nindm1) for therapeutic purposes. For example, brentuximabvedotin (sgn-35) containing MMAE was developed for the treatment of Hodgkin’s and anaplastic large cell lymphoma. As for Ado-Trastuzumab Emtansine, also known as T-dm1, the antibody trastuzumab (humanized IgG1 anti HER-2 antibody) is chemically linked to the drug maidenlignindm1. Noteworthily, once trastuzumab binds to the HER-2/neu receptor on target cells, it can also prevent homologous or heterodimerization of the receptor (HER2/HER3) and inhibit the activation of mitogen activated protein-kinase(MAPK) and PI3K/AKT signal pathway, finally preventing the growth of tumor cells. At the same time, T-dm1 can be internalized into cancer cells and then binds to tubulin to induce cell death [57,58]. Although efforts have been made to develop ADCs, studies have shown that the blood stability of the second generation ADCs is not ideal, showing unfavorable in vivo toxicity, such as hepatotoxicity, cardiotoxicity, peripheral neuropathy, thrombocytopenia, and ocular toxicity [59]. 

After the emergence of clinical problems related to the second generation of ADCs, the third generation of ADCs were designed and prepared. The main understanding of the third generation ADCs is to design immunoglobulin G (IgG) molecules with proper drug binding positions, so as to obtain more uniform drug conjugates. In the third generation of ADCs, the instability of ADC in circulation can be overcome by biocoupling chemistry. The main goal is to reduce the uncoupling of drugs in blood circulation for minimized off-target toxicity, improve the therapeutic index and expand the treatment window. Seattle Genetics has developed Vadastuximab talirine (SGN-CD33A), which contains a novel synthetic pyrrolobenzodiazepine (PBD) dimer. It was coupled to a humanized anti-CD33 IgG1 antibody through a maleimidocaproyl valine–alanine dipeptide linker and is structurally related to anthramycin, leading to targets cell death by cross-linking DNA and effectively preventing cell division. Vadastuximab talirine not only demonstrates robust activity in a series of acute myeloid leukaemia(AML) animal models but also overcomes transporter-mediated MDR [59,60,61,62]. Additionally, the better introduction of “cleavable” linkers between antibodies and payloads also promotes the clinical development of the third generation of ADCs, due to their strengths in controllable drug release at the target site. These cleavable linkers include acid-sensitive linkers (e.g., hydrazone linkage), reducible disulfides, and enzyme cleavable linkers, which have been comprehensively reviewed by Bargh and co-workers [63]. Taking reducible disulfide as an example, Pillow and co-workers directly attached maytansinoid bearing thiols (DM1) to engineered cysteine residues in an antibody. The obtained ADCs were stable during blood circulation due to the shielding of disulfides by antibodies. Once internalized by target cells followed by antibody catabolism, the disulfide linkers were exposed to the reducing cellular environment, leading to a rapid disulfide catabolite and desirable drug release [64]. Rémy Gébleux and co-workers also developed two ADCs by using a similar coupling strategy, denoted as SIP(F8)-SS-DM1 and IgG(F8)-SS-DM1. In this work, the F8 antibody, directed against the alternatively spliced Extra Domain A (EDA) domain of fibronectin was selected, but in two different formats which are IgG and small immune protein (SIP), respectively. Based on their results, IgG(F8)-SS-DM1 was more stable in mouse plasma than SIP(F8)-SS-DM1, demonstrating a novel mechanism in the drug release from the disulfide-based ADCs. However, the ADCs in SIP format displayed a better therapeutic outcome compared with ADCs in the IgG format in immunocompetent mice bearing F9 tumors, revealing that the format of antibodies plays a significant role in determining the final therapeutic efficacy [65]. Immunotoxins, that ADCs created by chemically conjugating antibodies to whole protein toxins (lack of natural binding domain), have been proved to enhance the performance of toxins by taking the advantage of the desired specificity of antibodies to the target cells and the potency of the toxins to kill cells effectively [66,67]. In this kind of ADC, the disulfide linkers could also be introduced to control the release of toxins. For instance, two immunotoxins (ITxs) were constructed via chemical conjugation (disulfide linker) of the ribosome-inactivating proteins (Saporin-S6) with anti-CD20 mAb Rituximab, with the difference in the structures (monomeric and dimeric) and molecular weight, known as HMW-ITx (dimeric) and LMW-ITx (monomeric), respectively. Accordingly, the HMW-ITx was more cytotoxic than the LMW-ITx in two CD20+ lymphoma cell lines, Raji and D430B, thanks to its higher toxin loading and more efficient antigen capping, although they displayed similar activity in inhibiting protein synthesis in a cell-free system. Moreover, as compared with parent Saporin-S6, both ITxs are more active [68].

The third generation ADCs have many other advantages, including improved stability, optimized pharmacokinetics, slow deconjugation, and high activity against cells that express lower levels of antigen. These ADCs are site-specific and homoconjugated, which provides a basis for applying ADCs in cancer treatment. However, many of these ADCs are still in the research stage, and only a relatively small number have been tested in the clinic. Although they can improve cancer treatment, there are still concerns about their limited treatment range [69,70,71,72]. More details about the potential of ADCs have been well reviewed elsewhere, along with their status in the oncology market [73]. At present, a total of 14 ADC drugs in the world have been approved for listing (Table 2). Among them, myLotarg of Pfizer/Wyeth is the first ADC drug to be listed, for the treatment of acute myeloid leukemia, but due to fatal liver injury from Mylotarg, Pfizer withdrew Mylotarg in 2010. In 2021, FDA successively approved the listing application of Zvnionta and Tivdak. The former was a CD19 ADC drug and the latter is a tissue factor (TF) ADC drug [41,74,75].

### 2.3. Peptide-Drug Conjugates

Peptides are short chains of amino acids, which are distinguished from proteins by their shorter length. Based on the definition from the FDA, a peptide is a polymer composed of less than 40 amino acids (500–5000 Da). Peptide–drug conjugates (PDCs) have a structure similar to that of ADCs, which also consist of three important components, including functional peptides, linkers and cytotoxic drugs. In 2021, Melflufen^®^ (melphalan flufenamide), as a first-in-class anticancer PDC, was approved by the FDA for the treatment of relapsed and refractory multiple myeloma (RRMM) in combination with dexamethasone. Once in tumor cells, the aminopeptidase fusion domain of Melflufen can be affected by aminopeptidase and lipase, and the hydrophilic alkylating agent melphalan is further released to inhibit the DNA reparation and angiogenesis during therapy. Moreover, a series of hydroxypeptidases expressed in multiple myeloma and other tumors can be cracked by a hydrolysis reaction and quickly released in the cytotoxic payload from Melflufen [76]. 

For some PDCs, the homing peptides are responsible for directing the entire PDCs to the targeted tumor cells by recognizing the specific receptors overexpressed on the cellular surface, with the purpose of decreasing the side effects from off-target delivery. These homing peptides include RGD (targeting integrins), GnRH (targeting gonadotropin releasing hormone receptor(GnRH-R)), SST (targeting somatostatin receptors (SSTR1–5)), EGF (targeting epidermal growth factor receptor (EGFR)), Angiopep-2 (targeting low-density lipoprotein receptor-related protein-1 (LRP-1)) and so on [77]. It has been reported that the secondary structure of these peptides pronouncedly influences their binding affinity. Moreover, there are still some limitations compromising their homing effects, such as fast degradation by enzymes at terminal sites, chemical instability and rapid renal clearance. To circumvent these problems, the techniques for cyclization and stapling of linear peptides have been developed with the details reviewed by Cooper et al. recently [78]. In addition to the linear peptides, peptide dendrimers have also been studied as building blocks in PDCs. The employment of peptide dendrimers is considered to be beneficial because of their adjustable amino acid characteristics and good biocompatibility. Recently, Oliveria and co-workers developed the peptide dendrimer–gemcitabine (GEM) conjugates for the treatment of colorectal cancer. YIGSR, a kind of peptide has been shown to selectively bind to laminin receptors (LR) overexpressed in many cancers. By conjugating YIGSR peptide with GEM, YIGSR-conjugated peptide dendrimers could be selectively internalized into HCT-116 cancer cells with high expression of LR [79]. Although peptides are currently underrepresented in clinical trials compared with small molecules and biological agents, they still provide excellent versatility and can help to design targeted therapies.

In addition to the peptides with the function of cell targeting, a series of cell penetrating peptides (CPP), that promote the drugs to enter cells through a non-specific mechanism, have also been prepared to develop the PDCs. CPPs usually have characteristic features, such as hydrophobicity, amphipathicity and net positive charge. CPPs-mediated cellular internalization is an energy-dependent cell process, such as endocytosis or receptor-mediated uptake [80,81]. Although CPPs can enhance the in vitro and in vivo efficacy of impenetrable molecules in biomedical applications, it still has the limitations of low osmotic concentration and poor target selectivity [82,83,84,85]. Furthermore, cationic CPPs present problems, such as their inability to selectively home to the target. To improve the performance of CPPs-based PDCs, recently, a study found that multimers of lysine (K) and leucine (L) of the amphiphilic α-helical LK sequence can penetrate cells at the nanomolar level, which was 100–1000 folds lower than the transverse concentration of traditional CPPs [86]. The stronger the interaction between CPPs and cell surface receptors, the faster the PDCs enter the tumor cells [87,88]. Similarly, other research groups have reported that amphiphilic CPP in the form of dimers showed stronger cell penetration activity than monomer CPP [89,90]. Amphiphilic cyclic cell-penetrating peptides (cCPPs) are a relatively new class of peptides. The cCPPs possess several advantages over the conventional linear CPPs, such as low immunogenicity, proteolytic resistance, improved cellular uptake, facilitated serum stability, and better interaction with the membrane receptors. Park and co-workers conjugated cabazitaxel (CBT) to a cCPP via an ester bond to assist CBT to penetrate into the tumoral cells. The conjugates showed less toxicity to normal human embryonic kidney (HEK-293) cells compared to free CBT while displaying approximately three- to four-fold higher antiproliferative activity on cancer cell lines, compared to the free CBT analog. Although the increased drug delivery can be attributed to the presence of CPPs, the widespread use of these types of PDCs is still limited due to their low cellular specificity [91]. 

Moreover, some peptides in PDCs can also be cleaved under various stimuli, acting as linkers for controllable drug release. Tripodi and co-workers conjugated daunorubicin with a cyclic peptide containing the NGR motif, which is a tripeptide sequence capable of recognizing CD13 receptor isoforms on tumor cells. An enzyme-cleavable tetrapeptide (GFLG) was selected to conjugate the cyclic peptide with daunorubicin. Cathepsin B overexpressed in the tumor microenvironment (TME) could cleave GFLG connectors to daunorubicin to avoid the undesired toxicity on normal tissues. Based on the results, the mice bearing subcutaneous Kaposi’s sarcoma can tolerate this conjugate and express plasma stability and antitumor activity both in vitro and in vivo. Compared with free daunorubicin, the PDC-based therapy indeed decreased toxic side effects and improved the efficacy of tumor growth inhibition in mice [92]. 

In another case, the peptides in PDCs could also participate in the assembly of nanoparticles, facilitating the entry of drugs into the nucleus, finally influencing the nuclear DNA or related enzymes for a therapeutic purpose. In detail, Cai and co-workers combined negatively charged 10-hydroxycamptothecin (HCPT) with peptide amphiphiles FFERGD to obtain HCPT–FFERGD (HP) firstly, followed by forming two kinds of complexes with positively charged cisplatin (complex 1 and 2) by adjusting the molar ratio of HP and cisplatin. Interestingly, complex 1 (HP/cisplatin = 1:1) and complex 2 (HP/cisplatin = 1:1.5) can self-assemble into nanostructures of different shapes, which were rod-shaped nanofibers for complex 1 and spherical nanoparticles for complex 2, respectively. These obtained nanostructures exhibited the following advantages: Firstly, HCPT and positively charged platinum cannot easily enter the cell membrane but can enter the cell membrane smoothly after coupling with a polypeptide. Secondly, both rod-shaped nanofibers and spherical nanoparticles can effectively enter the nucleus, which was not observed for the group of HCPT and HP-based nanoparticles. Thirdly, the results also showed that both rod-shaped nanofibers and spherical nanoparticles not only effectively inhibited cancer cells in vitro and in vivo but also enhanced the inhibitory effect on drug-resistant tumor cells. The team likened their nanomedicine to a “Trojan horse”, that transports soldiers (anticancer drugs) through the walls of the castle (the cell and nuclear membrane) for precise targeted and synergistic therapy (Figure 4) [93].

### 2.4. Drug-Drug Conjugates 

#### 2.4.1. Conjugates of Multiple Chemotherapeutic Drugs

Drug–drug conjugates present accurately defined chemical structures with a low molecular weight and high drug loading content (even up to 100%) as compared with other drug conjugates. They are usually connected by two anticancer drugs, which can self-assemble into nanoparticles without additional stabilizers. The formed nanoparticles can enhance drug accumulation in tumors through the EPR effect. Under certain stimuli, the structural integrity of nanoparticles could be damaged along with the rapid release of drugs for the synergistic chemo–chemo therapy. Compared with a single administration, the conjugates of these chemotherapeutic drugs showed excellent cytotoxicity and fewer side effects [75,94,95]. In some cases, two identical chemotherapeutic drugs are bound to the same active site through appropriate junctions to form the drug–drug conjugates, known as drug dimers. The self-assembly of drug dimers can improve the water solubility of hydrophobic drugs as well as their stability in physiological conditions. For example, Li and co-workers prepared a doxorubicin dimer (D-DoxCAR), which is synthesized via a carbamate linkage. It was further prepared into nanoparticles (D-DoxCAR-NP) with a drug content up to 86%. The acidic tumor environment induced the cleavage of carbamate linker to induce the rapid release of DOX, resulting in an enhanced anti-tumor effect on HepG2 cells compared with parent DOX [96]. 

When two therapeutic drugs with different aqueous solubility and therapeutic mechanisms are combined through a properly designed joint, the therapeutic drug–drug conjugations could be obtained for self-assembly into nanoparticles. The resulting nanoparticles showed enhanced therapeutic efficacy through synergistic therapy compared to drug administration alone [97]. For instance, Podophyllotoxin (PPT) displays a significant anticancer effect by destabilizing microtubules and preventing cell division. However, it has been associated with some limitations, such as poor aqueous solubility and severe off-target side effects. To overcome these disadvantages, Hou and co-workers conjugated hydrophobic PPT and hydrophilic methotrexate (MTX) via a reduction-responsive disulfide bond. The nanoprodrug formulated by this amphiphilic drug conjugates the successfully improved tumor delivery of PPT [98]. Huang and co-workers synthesized an amphiphilic drug–drug conjugate from the hydrophilic anticancer drug irinotecan (Ir) and hydrophobic anticancer drug chloramphenicol (Cb) through hydrolyzable ester ligation. Amphiphilic Ir-Cb then self-assembled into nanoparticles and achieved passive tumor targeting via the EPR effect. The ester bonds were further cleaved in tumor cells to realize the recovery of cytotoxicity of both drugs [99]. The biological applications of curcumin (CCM) are severely limited by some undesired properties, such as poor water solubility, short serum half-life, and low bioavailability. Cheng and co-workers utilized bifunctional PEG, bearing both azide and carboxylic acid groups, to realize the conjugation of Erlotinib (ELT) and CCM. Compared with free drugs, the conjugates prolonged the half-life of drugs in blood retention, and enhanced drug accumulation in tumor tissues (Figure 5A–C) [99].

#### 2.4.2. Photothermal Agent-Drug Conjugates

Photothermal therapy (PTT), as one type of phototherapy, uses the heat generated from photothermal agents (PTAs) upon near-infrared (NIR) laser irradiation to induce the death of cancer cells [100,101,102,103,104]. The PTAs should meet the requirements, including low cytotoxicity, easy preparation and modification, and good solubility in biocompatible liquids. Besides, the absorption of PTA is usually adjusted to the range between 750 and 1350 nm (Biological Windows (BWs), BW I: 700 nm–980 nm/BW II: 1000 nm–1350 nm) to enhance the penetration of light in the tissues. The ideal PTA should also have a higher photothermal conversion efficiency and good accumulation in tumor tissues. Nowadays, the PTAs can be divided into two categories, which are inorganic PTAs (e.g., metallic NPs, carbon-based NPs) and organic molecular-based ones, respectively, both of which could be prepared into drug conjugates for cancer treatment. In this section, we focus on the organic PTAs-related conjugation and the inorganic PTAs-related ones will be detailed and introduced in Section 2.5 “Inorganic Nanoparticle-Drug Conjugates”.

For organic molecular-based PTAs, their electrons absorb photon energy upon light illumination and transform from the ground state to the excited singlet state. When the electrons return to the ground state, due to the non-radiative relaxation processes in which excited singlet states collide with their neighboring molecules, photothermal effects could be induced. The increase in kinetic energy leads to heat surrounding the microenvironment and induces irreversible damage to tissues when the temperature exceeds the threshold [105]. Organic PTA includes organic dye molecules (e.g., indocyanine green and heptylcyanine) and organic nanoparticles (e.g., semiconductor polymer nanoparticles (SPNP)) [106,107]. PTT can target tumors precisely with adjustable doses of external laser irradiation, thus minimizing the damage to surrounding healthy tissue. It has been recognized as an effective non-invasive therapy available for most types of cancer [108]. However, there are still some obstacles during PTT, such as low transmission efficiency of PTA in tumors, and heat-induced overexpression of heat shock proteins (HSPs) in tumors post-PTT responsible for thermal resistance. Therefore, the combination of PTAs with anticancer drugs can potentially overcome these drawbacks of PTT in a synergistic manner [109,110,111,112]. Furthermore, PTAs–drug conjugates related nanomedicines continue to present the advantages of drug conjugates for an improved therapeutic outcome. For example, in order to improve the transfer efficiency of photothermal agent IR820 and the chemotherapeutic drug PTX, Zhang and co-workers coupled IR820 with PTX to form pH and enzyme-sensitive carrier-free nanopharmaceuticals, which were used for fluorescence imaging-guided synergistic chemotherapy-PTT. The nanosystems show high drug loading content and promising stability. IR820-PTX also solved the problems of the short life span of IR820 in vivo and poor solubility of PTX and effectively inhibited tumor growth via combined PTT and chemotherapy [112]. Similarly, Ao and co-workers linked camptothecin (CPT) to IR820 via a redox disulfide ligand to form the PTA-chemotherapeutic drug conjugate IR820-SS-CPT. The drug load content of IR820-SS-CPT in nanoparticles formed by self-assembly was close to 100%, and the water solubility of CPT and the membrane permeability of IR820 were significantly higher than that of a single drug [113]. Du and co-workers designed a self-assembled vector-free nanomachine (IR820/ATO NPs) based on ATovaquone (ATO) and IR820, which successfully addressed the problem of hydrophilic IR820 being unstable and easily removed in vivo. More interestingly, ATO can act as an oxidative phosphorylation system (OXPHOS) inhibitor to inhibit mitochondrial respiration and reduce the synthesis of ATP after entering tumor cells, thus leading to the downregulation of HSPs and synergistic enhancement of the sensitivity of PTT treatment in tumor cells [114].

#### 2.4.3. Photosensitizer-Drug Conjugates

PDT is another important type of phototherapy, during which the photosensitizers (PSs) in target lesions are stimulated by light sources at a specific wavelength to produce reactive oxygen species (ROS), finally resulting in cell apoptosis and necrosis [97,115,116,117]. Upon light irradiation, PSs can be excited from the singlet basic energy state S_0_ to the excited singlet state S_1_ first, followed by partially transforming to the long-lived excited triplet state T_1_ through the intersystem crossing, which is the therapeutic form of the PSs. Subsequently, there are two mechanisms for ROS generation at the present stage: in the type I pathway, the excited PSs participate in the electron transfer reaction to produce free radicals and free radical ions. In the type II pathway, the PSs transfer energy to molecular oxygen (^3^O_2_) to produce highly reactive singlet oxygen (^1^O_2_). It needs to be mentioned that type II-based PDT is more popular than type I-based PDT [118]. Thereafter, the generated ROS can trigger oxidative stress on tumor cells, leading to the activation of the protein kinase pathway, expression of transcription factors and cytokines, and release of factors mediating apoptosis, resulting in the apoptosis or necrosis of tumor cells. Furthermore, PDT can effectively target tumor blood vessels, causing damage to the tumor vasculature, resulting in the injury of vascular endothelial cells, disorders of endothelial structure, and a significant reduction in the tumor cell nutrition supply. PDT has also been reported to induce acute local and systemic inflammatory responses, ultimately stimulating T cell activation and generating anti-tumor immune responses [119,120,121,122].

As for PSs, they can be divided into non-porphyrin PSs (e.g., rose red (RB) and methylene blue (MB), ruthenium (II) complexes, fullerenes, etc.) and porphyrin PSs (such as porphyrin, phthalocyanine (Pc), naphthalene phthalocyanine (Nc), etc.) [105]. In recent years, some photosensitive compounds extracted from herbal plants, such as chlorophyll and curcumin, have proven to possess photodynamic activity, which can also be used as PSs in PDT. Compared with other routinely used PSs, natural PSs usually causes lower side effects. Despite the success of PDT, new PSs and innovative methods are still needed to improve the practical application of PDT in clinical oncology. More and more studies have reported that the combination of PSs and nanomaterials, such as photosensitizer-drug conjugates-derived nanoparticles and PS-antibody conjugates [123], can improve the efficiency of PDT and diminish the undesirable side effects. Hao and co-workers formed CPT-TK-HPPH nanoparticles by a photosensitizer-drug conjugate, that combined CPT with PS 2-(1-hexoxyethyl)-2-dvinyl pyropheophorbide-a (HPPH) via a ROS-responsive thioketal bond. The platinum was then loaded into CPT-TK-HPPH to produce CPT-TK-HPPH/PT nanoparticles. The resultant nanoparticles can efficiently catalyze hydrogen peroxide (H_2_O_2_) to produce oxygen (O_2_), realizing tumor oxygenation for improved HPPH-mediated PDT. Moreover, the generated ROS could further induce the cleavage of the thioketal linker for the release of CPT on demand. The CPT–TK–HPPH/PT NP effectively inhibited colon tumor proliferation and growth in vitro and in vivo [124]. In another example, Ha and co-workers combined the chemo-drug combretastatin A-4 (CA4) with a tumor-targeting biotin portion and a PS Zn (II) phthalocyanine (ZnPc), in which a ROS-sensitive aminoacrylate linker was introduced for the controlled release of CA4 during PDT [125]. Besides ROS-responsive linkers, Um and co-workers used caspase 3 cleavable peptide (Asp-Glu-Val-Asp, DEVD) as a linker for the conjugation of Ce6 (PS) and MMAE (anti-cancer drug) and developed the Ce6-DEVD-MMAE nanoparticles. Compared with traditional PDT using high-energy irradiation, the new therapeutic strategy used lower-energy irradiation to induce the apoptosis of cancer cells. Along with MMAE-mediated anticancer activity, strong cytotoxic effects could be induced upon exposure to lower-energy irradiation. More importantly, Ce6–DEVD–MMAE nanoparticles did not display any toxicity in the absence of light illumination due to the drug conjugation strategy, which was different from free MMAE (1–10 nM) which had obvious cytotoxicity (Figure 5D,E) [126].

Additionally, recombinant antibody fragments have also been conjugated with PSs to realize antibody-directed PDT, of which the structure is similar to ADCs. These antibody–PS conjugates present promising strengths in superior drug loading, more favorable pharmacokinetics, enhanced potency and target cell selectivity [123]. For example, Ebaston and co-workers successfully conjugated trastuzumab (Ab, targeting Her2 receptors) with mI_2_XCy(PS). Interestingly, the hydroxyl group in mI_2_XCy was further protected by acetyl (Ac) to quench the ROS generation and fluorescence emission of PS, with the purpose of reducing the side effects caused by existing PSs to organs due to insufficient specificity. Upon the Ac group being cleaved by the intracellular esterases, the photodynamic activity could be restored and effective ROS generation could be observed upon NIR light irradiation. As desired, these Ab-mI_2_XCy-Ac conjugates displayed negligible side effects and promising tumor growth inhibition in the Her2 positive BT-474 tumor mouse model, which is almost the same as for the permanently active antibody–PS conjugates (Ab-mI2XCy) [127].

### 2.5. Inorganic Nanoparticle-Drug Conjugates

Among the various categories of nanocarriers, inorganic material-based ones are a matter of huge interest in developing nanoplatforms for cancer diagnosis and treatment, due to their excellent physical and chemical properties in the aspect of magnetic, thermal, optical, and catalytic performance. Moreover, these inorganic nanoplatforms are able to encapsulate and release drugs in a controllable manner, extend the systemic circulation, decrease the undesirable side effects, and improve biocompatibility and pharmacological profiles. Broadly, the inorganic nanoparticles could be divided into two different categories, which are metallic nanoparticles and nonmetallic nanoparticles, respectively [126].

Bulk metal-related electronic sensors have been widely applied in the field of disease diagnosis. Interestingly, the post-processing of metals into nanoparticles endows them with special features, rendering them good candidates for biomedical application. The common metallic nanostructures comprise gold nanoparticles, silver nanoparticles, iron oxide nanoparticles, calcium nanoparticles and so on. They are capable of both active and passive targeting, finally obtaining promising flexibility. Furthermore, the metallic nanoparticles themselves also display therapeutic functions during cancer treatment. For example, gold nanoparticles (AuNPs), which could be subdivided into gold nanospheres, gold nanorods, gold nanoshells, gold nanoclusters, etc., have shown potential applications in PTT and radiofrequency ablation [128]. Silver nanoparticles (AgNPs) present several intrinsic anticancer functions, including halting the generation of ROS by influencing the cellular mitochondrial system, reducing the velocity of ATP synthesis, and altering the corresponding biological pathways necessary for the survival of tumor cells [129]. Iron nanoparticles, such as iron oxide nanoparticles (IONPs) and superparamagnetic iron oxide nanoparticles (SPIONs), are indispensable conditions to realize magnetic hypothermia treatment (MHT). More importantly, the drugs can be covalently connected to these metallic nanoparticles through the chemical reaction between different functional groups, such as thiol/sulfide, carboxylic/amine, azide/alkyne and so on. Therefore, the therapeutic index and the pharmacokinetics of drugs can be significantly improved compared with native drugs [130,131,132]. For example, AgNPs were chemically conjugated with the somatostatin analog octreotide (OCT) through amide bonds to form AgNPs–OCT by Abdellatif and co-workers, which was further combined with alginate to produce AgNPs–OCT–Alg. It was found that AgNPs–OCT–Alg can not only effectively accumulate in lung tissue with promising delivery applicability and cytotoxicity, but also reduce drug side effects [133]. Gold nanorods (AuNRs) are good candidates as PTAs for PTT due to two different kinds of surface plasmon resonance (SPR), in which the strong longitudinal mode can be adjusted to visible light and the NIR region [134,135,136]. Jongseon and co-workers prepared AuNRs with different aspect ratios as PTAs to conjugate folic acid (FA)–PEG monoblock copolymer (FAP) and pheophorbide a (Pheo), yielding nanoplatform denoted as FAPAuNR–Pheos. The nanoplatform exhibited excellent performance in singlet oxygen generation, photothermal conversion, and glutathione(GSH)-responsive release of Pheo. Furthermore, FAPAuNR–Pheo with tumor-targeting FA ligands exhibit promising tumor-targeting activity and a synergistic PTT/PDT effect (Figure 6) [137].

Nonmetallic nanoparticles commonly refer to nanostructures based on silicon and carbon materials. These nanoparticles could be mesoporous, represented by the mesoporous silica nanoparticles (MSNs), which offer them promising properties, such as high drug loading capacity and pleasant surface chemical modification. These characteristics promote a great potential for smart drug delivery. MSNs are traversed by pores with a nanometer scale and are usually synthesized through a top-down approach with the assistance of chemicals and metals [138]. Due to the special porous structures, they present high drug loading ability and diverse surface functionalization. Moreover, the particle sizes, pore sizes and particle shape are adjustable by changing the precursor (e.g., tetraethyl orthosilicate (TEOS)) concentrations and stirring conditions. In order to increase the drug loading content and loading stability, the drugs can also be retained in the silica network through chemical conjugation (on the surface of MSNs or inside the pores) and released in a controllable manner. In addition, MSNs are also biodegradable. Their decomposition products, orthosilicic acid, are harmless to the health, which is one of its advantages over other inorganic nanoparticles [138,139]. However, exposed monodisperse silica microspheres tend to exhibit high aggregation when directly exposed to the biological environment, limiting their application in the biomedical field which could be overcome by the proper surface modification on mesoporous silica. Peng and co-workers designed a nanoplatform based on MSN for synergistic PTT/chemotherapy. The polymer poly(PEGMA-co-HEMA) was firstly modified onto the surface of MSNs via surface-initiated atom transfer radical polymerization. Then, DOX was anchored onto the polymer via reversible covalent bond cis-aconitic anhydride with pH sensitivity. Indocyanine green (ICG), as PTA, was further loaded into the pores of MSN for PTT, obtaining MSN@poly(PEGMA-co-HEMA-g-DOX)/ICG. This nanoplatform presents an improved synergistic effect on both HepG2 and Hela cells by virtue of photothermal action and promoted linker cleavage [140].

Carbon nanoparticles applied in drug delivery are usually sp^2^ carbon materials, including single-walled carbon nanotubes (SWCNTs) and multiwalled carbon nanotubes (MWCNTs), graphene, fullerene and carbon dots. They possess unparallel advantages, such as easy surface modification, strong adsorption, high photothermal conversion efficiency, supramolecular π–π stacking and excellent biocompatibility. Actually, therapeutics could also be covalently functionalized onto the carbon nanoparticles, since the carbon materials could bear the carboxylic acid groups once undergoing treatment with strong acid solutions, which provides a reaction site for the drug conjugation. Besides, carbon materials could also be functionalized via 1,3-dipolar cycloaddition of azomethine ylides, which provide carbon nanoparticles with customized substituents relying on the structure of α-amino acids and aldehydes employed during modification [141]. Dhar and co-workers prepared an axial folate derivative (FA)-containing platinum(IV) complex and further conjugated it with SWCNTs. This platinum-single-walled carbon nanotube structure containing folic acid has the targeting ability, which increases the activity of the platinum-based anticancer drug and significantly enhances the cell killing performance [142]. Recently, quantum dots (QDs) and ceramic-based nanoparticles have been paid more attention to due to their anticancer potential [143,144].

## 3. Representative Applications of Conjugated Nanomedicine

Chemotherapy is one of the standard methods for the clinical treatment of malignant tumors. Due to the heterogeneity of tumors and the complexity of their pathological mechanisms, a single chemotherapeutic drug is usually unable to eradicate cancer cells. It may also encounter some problems, such as toxic side effects induced by high doses of drugs and obtaining MDR after repeated treatment. These problems then increase the likelihood of cancer metastasis or recurrence [145,146,147]. The emergence of the combination of multiple antineoplastic drugs makes up for the deficiency of single drug application. Accordingly, the overall treatment benefit of the multidrug combination is usually higher than that of single drug administration by virtue of different therapeutic mechanisms. More importantly, the drug dose used during synergistic therapy usually decreases and the unfavorable side effects could be weakened under the premise of the same or better therapeutic efficacy [148].

Nevertheless, the traditional “cocktail” therapeutic strategies also display limitations. The actual concentration of individual drugs in tumors is uncontrollable, mainly due to differences between drugs in physical and chemical properties, pharmacokinetics, tissue distribution and so on. Since the drug ratios play a significant role in their synergistic manner, the final therapeutic outcome cannot be guaranteed during “cocktail” treatment. In contrast, conjugated nanomedicine usually presents a relatively fixed drug ratio in blood and tumor tissues depending on the initial composition of the materials. The stable drug connection also prevents the undesired pre-mature drug release, further promoting the accurate delivery of multiple drugs to targets as well as promising therapeutic efficacy. In this section, we will introduce some representative applications of conjugated nanomedicine in synergistic chemotherapy.

### 3.1. Synergistic Chemo-Chemo Therapy

Different types of chemotherapeutic drugs can be classified according to their function and mechanism of action on cancer cells. The most common classes of chemotherapeutic drugs are alkylating agents, antimetabolic agents, anthracyclines, topoisomerase inhibitors, mitotic inhibitors, and corticosteroids. The choice of two or more chemotherapeutic drugs depends on the stage and type of cancer, the synergistic behaviors of various drugs and other factors. The choice of drugs determines whether the effect is synergistic, additive, or antagonistic [149]. Conventional “cocktail” treatment generally presents an inadequate enhancement of therapeutic efficiency since different anticancer drugs often display diverse pharmacokinetics and transmembrane ability, resulting in the uncontrollable distribution of drugs in tumors [150,151,152]. To overcome the above challenges, nano-carrier based multiple drugs co-delivery systems have been developed for cancer synergistic chemo–chemo therapy, which are capable of delivering two or more chemotherapeutics to tumors in a desirable ratio by virtue of the advantages of nanomedicine. Once connected, the hydrophobic and hydrophilic anti-tumor drugs, serving as building blocks, could also provide the impetus to fabricate the nanoparticles. This kind of conjugated nanomedicine not only displays the advantages of high drug loading content but also relatively stable drug delivery capacity, reduced side effects and enhanced anticancer activity benefitting from the synergistic effect of different drugs [153,154,155]. Previously, we have introduced the conjugated nanomedicine (denoted as MTX-SS-PPT NAs) developed by Hou and co-workers, which is self-assembled from the drug conjugates containing hydrophilic drug MTX and the hydrophobic drug PPT. In this nanomedicine, the disulfide bonds connecting the two drugs contribute to the degradation of MTX-SS-PPT NAs in tumor cells under reduction conditions. As a slow-release of the active drug, the nanoagent can significantly improve the biocompatibility of PPT and reduce its toxicity [98]. 

The development and metastasis of solid tumors highly depend on the formation of neovascularization. However, the use of angiogenesis inhibitors alone cannot meet the needs of cancer treatment. Sun and co-workers conjugated hydrophilic chemotherapeutic drugs (fluorosarboside, FUDR) with hydrophobic antiangiogenic drugs (pseudoboric acid B, PAB), followed by formulating nanoparticles in an aqueous solution. These nanoparticles not only displayed promising anti-tumor activity but also had efficient antiangiogenesis properties, leading to a good cancer therapeutic outcome in mice bearing subcutaneous HeLa tumors [156]. Dasatinib (DAS) is a competitive oral dual Src/Abl kinase inhibitor, which can inhibit a variety of Src signal pathways and further inhibit tumor cell migration, invasion and angiogenesis [157]. Yang and co-workers linked DAS with cisplatin octahedral coordination derivative diamino dichlorodihydroxyplatinum (DH-CP) through an ester bond to form an amphiphilic drug–drug conjugate (CP–DDA) at the ratio of 2:1. Then, the stable nanoparticles (CP–DDA NPs) were formed by the self-assembly of CP–DDA in an aqueous solution. The nanoparticles displayed promising stability during blood circulation and increased accumulation of drugs in the tumor site through the EPR effect. After being internalized by cancer cells, under the action of high GSH and esterase, the DAS and CP could be released in situ for inhibiting Src activity and inducing cell apoptosis, respectively, resulting in a synergistic anti-tumor effect [158].

### 3.2. Synergistic PDT/PTT-Chemo Therapy

Phototherapy, including PDT and PTT, is a tumor resection and function-preserving interventional therapy, which shows great potential in clinical application. In the process of phototherapy, non-toxic phototherapeutic agents (PSs or PTAs) can be activated upon light irradiation, thus inducing cell death without causing undesirable collateral damage to normal tissue. However, it is difficult to completely eradicate solid tumors with single phototherapy. It has been reported that the combination of PTT and/or PDT with chemotherapy can provide therapeutic advantages including (1) giving play to synergistic effects during treatment. (2) Decreasing the undesirable side effects from anticancer drugs via lowering the drug dosage. (3) Facilitating the deep tumor penetration of chemotherapeutic drugs under hyperthermia treatment (PTT). (4) Promoting the cellular internalization of drugs in the presence of a large amount of ROS (PDT). (5) Inducing an immune response by phototherapy, including innate/adaptive immunity and antitumor immunity, to maximize the therapeutic outcome [105,159]. The unfavorable pharmacokinetic properties and desynchrony in the tumor accumulation of chemotherapeutic drugs and phototherapeutic agents still hinder the success of synergistic PDT/PTT-chemotherapy. Therefore, phototherapeutic nanomedicine has also aroused great interest in order to continuously improve its performance [97,160], among which conjugated nanomedicine occupies an important position [161,162]. As for synergistic PDT-chemotherapy, for example, Chen and co-workers developed a supramolecular system with optimized PS (4,4-difluoro-boradiazaindacene, BODIPY) and anti-cancer drug (PTX) loading efficiency. The adamantyl BODIPY (Ada-BODIPY) and PTX (Ada-PTX) were connected to the block copolymer (PEG-PGA-β-CD) through the host–guest interaction between adamantane and β-cyclodextrins (β-CD), followed by being prepared into nanoparticles of Ada-PTX (60%)-BODIPY(40%)-PNS. These nanoparticles remained in the precise drug loading ratio during circulation with the minimized pre-mature release of drugs. Upon NIR laser irradiation, ROS could be generated efficiently for PDT as well as the cleavage of ROS-sensitive aminoacrylate linker in Ada-PTX. Finally, PDT and cascaded Ada-PTX activation showed a significant inhibitory effect on tumor growth [163]. A similar nanosystem (PheoA-SN38-HC) has been developed by Lee and co-workers, which contains the ROS-cleavable thioketal-SN38 for the drug release during PDT, showing good tumor targeting for CD44 positive cancer cells and effective tumor inhibition mediated by synergistic PDT-chemotherapy (Figure 7A–C) [164]. 

Hypoxia is a key feature of the solid tumor microenvironment (TME) resulting from rapid malignant cell proliferation and vascular deformation during tumor angiogenesis, which is not beneficial to PDT. To overcome this problem, Xu and co-workers designed a nanoplatform self-assembled from amphiphilic oligomer Ce6-PEG Platinum(IV) (Ce6-PEG-Pt(IV), CPP) with upconversion nanoparticles (UCNPs) in the hydrophobic core. In this system, platinum(IV) diazido complexes bearing cis-diamine ligands can be activated to produce O_2_ as well as cytotoxic Pt(II) simultaneously upon laser illumination, successfully compensating for the consumption of O_2_ during the PDT process. The released active platinum(II) could also trigger efficient chemotherapeutic effects, resulting in dramatically enhanced synergistic PDT-chemotherapy [165].

As for synergistic PTT-chemotherapy, Li and co-workers designed a prodrug–hemicyanine conjugate (Cy-azo) based nanoplatform to achieve the combination of H-aggregation-improved PTT and sequential hypoxia-activated chemotherapy. In Cy-azo, nitrogen mustard was introduced into the NIR fluorescent group heptamethyl cyanamide through an azo bond, and the superposition of the conjugates promoted H aggregates, showing higher photothermal conversion efficiency than traditional cyanine dyes. In addition, under hypoxic conditions, the nitrogen mustard can be activated due to the cleavage of azo bonds and released in the hypoxic TME to induce cell death, thereby greatly reducing the side effects of chemotherapy [166]. In another work, Zhou and co-workers prepared the dual drugs-conjugated polydopamine nanoparticles (PDOXCBs) through the one-pot aqueous copolymerization of two dopamine prodrugs, which combined the NIR-mediated PTT with cocktail chemotherapy into one nanoplatform. Upon NIR irradiation, PDOXCBs presented a dramatic photothermal effect with the assistance of polydopamine nanoparticles as PTAs. Meanwhile, chemotherapeutic drugs, including DOX and chlorambucil (CB), could be released from the nanoplatforms under the stimuli of pH 5.0 and the reduced environment, respectively. The synergistic PTT-chemotherapy based on PDOXCB27 upon NIR irradiation displayed a highly lowered IC_50_ value on MCF-7 cells and a combination index of 0.36, revealing a promising combination between PTT and cocktail chemotherapy (Figure 7D,E) [167].

### 3.3. Synergistic Immune-Chemo Therapy

In recent years, with the development of molecular biology and tumor biology, tumor immunotherapy has become a new treatment method with a good application prospect. During cancer immunotherapy, the collective immune system can be activated by strengthening the natural immune defense of patients to fight against cancer cells and relieving the immunosuppressive microenvironment, to eradicate tumors and inhibit tumor metastasis and recovery. On the one hand, immunotherapy is aimed at training immune cells to recognize and remove target cells carrying tumor antigens, and enhancing immune-mediated tumor cell lysis, displaying the advantages of good curative effect, fewer adverse reactions and the prevention of recurrence. On the other hand, the down-regulation of the immunosuppressive signal pathways in tumor tissues could also facilitate the final immunotherapeutic effects [168]. So far, there have been several types of immunotherapy achieving great success in tumor therapy, such as immune checkpoint blockade (ICB), adoptive T cell transfer, cytokine therapy, agonist immunotherapy, vaccines and so on. However, due to the complexity and heterogeneity of tumors, systemic defects, such as immune escape and immunotoxicity of tumors make the overall efficacy of immunotherapy only about 20%. It needs to further improve the efficiency of tumor immunotherapy via inhibiting immune escape and enhancing the immunotherapeutic response rate. Among the various therapeutic strategies to improve the efficacy of immunotherapy, drug conjugates have been recognized as one of the good choices. Nano-drug delivery systems can enhance the retention, accumulation, penetration and target cell uptake of tumor immunotherapeutic drugs in tumor sites [169,170].

Besides, the combination of immunotherapy with other therapeutic modalities, such as chemotherapy and phototherapy, increases the immunotherapeutic effects. It has been recognized that chemotherapy can induce immunogenic cell death (ICD) to express or release damage-associated molecular patterns (DAMPs), including calreticulin (CRT), high-mobility group box 1 (HMGB-1), and adenosine triphosphate (ATP). These DAMPs are capable of enhancing the immunogenicity of cancer cells and stimulating the immune system to fight against tumors. Given this, Geng and co-workers developed the aptamer-drug conjugate nanomicelles to facilitate the antitumor immune response via DOX-mediated chemotherapy. In detail, an amphiphilic telodendrimer (ApMDC) consisting of an aptamer AS1411 and a monodendron connected with four DOX through acid-labile hydrazone spacers was firstly synthesized, followed by co-self-assembly with an ApMDC analog, in which the aptamer is substituted by PEG. Based on their results, the optimized micelles could induce ICD. Besides, the chemotherapy also promoted the tumor-specific immune responses of anti-PD-1 therapy (Figure 8A,B) [171]. In another work, Hu and co-workers developed a ROS-sensitive nanosystem (denoted as pep-PAPM@PTX) for synergistic chemotherapy and ICB therapy. The PD-L1-targeting D-peptide (NYSKPTDRQYHF, pep) was conjugated to the carrier materials and exposed to the surface of micelles with the function of anti-PD-L1 therapy. Accordingly, this micelle could bind to PD-L1 on the cell surface and promote its uptake via the lysosome-involved internalization, thus inhibiting PTX-activated PD-L1 upregulation and downregulating PD-L1 expression. It dramatically facilitated T cell infiltration and enhanced tumor immune activation, finally synergizing with PTX-mediated chemotherapy to achieve promising anticancer effects against triple-negative breast cancer (TNBC) [172]. Bai and co-workers designed a GSH/pH dual-responsive prodrug nano-platform (known as DDA) for synergistic chemotherapy/PDT/immunotherapy. The nano-platform can effectively enhance the immune response by promoting the maturation of dendritic cells and reducing the number of immunosuppressive immune cells, showing the enhanced adjuvant effect of anti-PD-1 therapy [173].

In addition to be applied for synergistic immune-chemo therapy, conjugated nanomedicine serves as a promising tool, and can also enhance the therapeutic efficacy of immunotherapy alone or other forms of synergistic immunotherapy, which is necessary to be discussed in this section. The company named Cyrtlmmune Sciences has developed a conjugated nanomedicine-related antitumor drug CYT-6091 (trade name: Aurimune^TM^) for cytokine immunotherapy. As mentioned in the previous section, gold nanoparticles can serve as PTAs and drug carriers simultaneously. The CYT-6091 was synthesized by covalent binding with recombinant human tumor necrosis factor (rhTNF) onto colloidal gold nanoparticles coated with mercapto functionalized PEG. It can be specifically stored in tumor tissues and has no obvious toxic and side effects [174]. It also provided the potential to combine with gold nanoparticles-mediated thermal therapy.

More recently, Xue and co-workers reported that the CD73 enzyme was highly expressed in tumor cells and immunosuppressive cells, including regulatory T cells (T_reg_ cells), myeloid suppressor cells (MDSCs) and M2-like tumor-associated macrophages (TAM.M2). However, CD73 was negative for non-immunosuppressive cells, known as lytic T lymphocytes, natural killer cells (NK cells) and dendritic cells (DC cells). Based on these findings, they developed an IR-700 dye-coupled anti-CD73 antibody (α-CD73-Dye), which could bind to CD73^+^ cells selectively. Upon NIR laser irradiation, these conjugates could perform photoimmunotherapy against targeted cells and prevent tumors from acquiring resistance to ICB, finally leading to advanced tumor eradication [175]. 

Another effective cancer immunotherapeutic modality is chimeric antigen receptor T-cell immunotherapy (CAR-T therapy), which is an adoptive T cell metastasis therapy (adaptive T cell metastasis, ACT), which infuses the patient’s T cells back into the patient to fight cancer. Compared with ordinary T cells, CAR-T is not restricted by the major histocompatibility complex (MHC), thus avoiding the immune escape of tumor cells with the low expression of MHC molecules on their surface. However, the immunosuppressive tumor microenvironment inhibits the infiltration of T cells, limiting the effect of CAR-T therapy. Luo and co-workers combined human serum albumin (€) and IL-12 into nanoparticles, which were further modified onto the surface of CAR-T cells via bioorthogonal chemistry to yield IL-12 nonstimulant engineering CAR T cells (INS-CAR T) hybrids. The IL-12 released from nanoplatforms can promote the secretion of CCL_5_, CCL_2_ and CXCL_10_, thus increasing the infiltration of CD8+ CAR T cells, relieving the immunosuppressive TME. Based on their results, the anti-tumor ability of CAR-T cells has been improved and the growth of solid tumors was inhibited with negligible side effects (Figure 8C–E) [176].

### 3.4. Synergistic PTT-TDT

As the prominent character of solid tumors, hypoxia impedes the therapeutic effect of oxygen-dependent radical-based cancer therapy, such as PDT and radiotherapy. To address these hypoxia issues, researchers have developed another type of oxygen-independent radical-based cancer treatment strategy, known as TDT. During TDT, the alkyl radicals can be produced upon heating with high efficiency due to the presence of thermally decomposable radical initiators, such serving as radical donors. More importantly, the aforementioned PTT could also serve as a heat source to induce the generation of alkyl radicals. Upon NIR light irradiation, light-triggered heat and heat-caused alkyl radicals can jointly damage vital cellular components and further induce cell death [177,178]. To improve the accumulation of radical initiators and PTAs in tumors and avoid the undesired pre-mature release of them during blood circulation, the PTA-initiator conjugated nanomedicine has also been developed for enhanced synergistic PTT-TDT. Xia and co-workers conjugated the photothermal PSs (porphyrin) with radical initiator 2,2′-azobis [2-(2-imidazolinI-2-yl) propane dihydrochloride (AIBI) and prepared the nanoparticles (tripolyphosphate (TPP)-NN NPs) with the assistance of pluronic F-127 as surfactant. The aggregated porphyrin could generate heat upon 638 nm laser illumination and then trigger initiator AIBI to produce alkyl radicals to induce cell death even in a hypoxia environment. TPP-NN NPs have shown the potential to inhibit the growth of cervical tumors without notable systemic toxicity [179].

In our previous study, an all-organic nanoparticle (denoted as ZPA@HA-ACVA-AZ NBs) realized the “precise strike” of hypoxic tumors via synergistic PTT/TDT. The loading strategy of radical initiators (ACVA) was optimized by the conjugation of alkyl chain-functionalized initiators ACVA-HDA to HA, thus averting the unfavorable adverse effect in normal tissues while improving the efficiency for the targeted delivery of radical initiators to solid tumors. Then, this amphiphilic hyaluronic acid (HA)-based lipoid (HA-ACVA-AZ) was used as a carrier to encapsulate the special zinc(II) phthalocyanine aggregates (ZPA), acting as PTAs for highly efficient PTT upon 808 nm laser irradiation. Therefore, the sequentially generated heat and alkyl radicals could simultaneously trigger cell death and restrain cancer metastasis under the action of PTT/TDT and CA IX inhibition [180]. Our group also developed carrier-free nano-theranostic agents (denoted as AIBME@IR780-APM NPS) for magnetic resonance imaging (MRI)-guided synergistic PTT/TDT. As an extension of previous work, we were devoted to the incorporation of diagnostic functions into the nanoplatform to improve the accuracy of synergistic therapy. As for the subject of this nanomedicine, the first IR780 derivative, IR780-ATU, was designed to conjugate the chelating agents (acylthiourea, ATU) with PTA with the purpose of chelating transition metal Mn^2+^ ions to perform the T1-weighted contrast-enhanced MRI. The other derivative, IR780-PEG, renders a nanosystem with high sterical stability, and increased solubility of hydrophobic IR780/dimethyl 2,2′-azobis(2-methylpropionate) (AIBME, radical initiator) and reduced risk from reticuloendothelial system (RES) uptake. Upon IR780-mediated PTT launched under NIR laser irradiation, AIBME could generate highly cytotoxic alkyl radicals, combing the heat from PTT to synergistically induce cell death, ignoring tumor hypoxia [181].

## 4. Challenges and Future Perspectives of Bioconjugation and Nanomedicine

Although conjugated nanomedicines have made significant progress in the treatment of tumors, they have shown great potential and application prospects, especially in protecting the loaded components, increasing drug selective accumulation and intratumoral penetration in tumor tissues, and reducing serious side effects in normal tissues. However, there are still many factors influencing the therapeutic efficiency and further clinical application of conjugated nanomedicine, such as the complexity of tumor biology, the biological interaction between the nanomedicine and the in vivo biological substances, and the large-scale production.

At present, more than 200 nanomedicines are in clinical research. Based on the reported data, although 95% of nanomedicines could pass the phase I clinical test, only 48% and 14% of these nanomedicines achieved good performance in phase II and III clinical tests. The major reasons causing the clinical failure of these nanomedicines are concluded as follows: 1. Biological barrier. Nanomedicines enter the body through a complex biological process to exert their efficacy. For example, before exerting their anticancer effects, nanomedicine should circulate in the body, accumulate in the tumor, penetrate into the deeper region of the tumor, internalize into tumor tissue and release the drugs. Every step in this process plays an essential role in achieving the desired therapeutic efficiency of nanomedicine. Thus, how to overcome these biological barriers during these steps critically influences their efficiency. 2. The linker in the conjugated nanomedicine significantly influences the targeted- and controlled-release of therapeutics. Thus, more intelligent and sensitive linkers should be explored and applied to achieve the “precise attack” of tumors. 3. Large-scale production. Large-scale production is also one of the major obstacles for nanomedicine transferring from bench to bedside. The process of large-scale can affect the physical and chemical characteristics of nanomedicine, such as the particle size, surface properties and drug loading capacity, which are vital for the biological effects of nanoparticles. Moreover, the toxic solvent usage during large-scale preparation may also influence the stability of materials or antibodies in conjugated nanomedicine. 4. Non-selective distribution. Although the targeted nanomedicine can accumulate in tumor tissues through the EPR effect and active targeting capacity in animal models, only 0.7% of the injected nanomedicine can actually be detected in tumors in the human body. Thus, it is still necessary to further explore the new mechanism for efficient targeted drug delivery. Thus, more specific antibodies and targeting strategies should be established to improve the selective distribution of conjugated nanomedicine. 5. The correlation between preclinical research and clinical trial results is not strong. Because the therapeutic efficacy of nanomedicine depends on its pharmacokinetics, tissue distribution, tumor accumulation, penetration, drug release, etc., the above characteristics are very different between animal models and patients. Furthermore, there is huge heterogeneity among different patients and tumor types. Therefore, improving the clinical therapeutic efficacy and comprehensive benefits is the key to promoting the clinical transformation of anti-tumor nanomedicines.

Lessons learned from completed, ongoing, or terminated clinical trials can help guide the forward development of nanomedicine. Clinical cancer treatment plans should refer to the cancerous tumors’ size, stage, location, and grade, thus the new generation of conjugated nanomedicines should be optimized and standardized in many aspects of clinical research, including patient screening, proper drug selection and proper combination with other therapies to further accelerate the development of conjugated nanomedicines.

Overall, through changing the strategies of many therapeutics’ applications, conjugated nanomedicine as one major branch of nanomedicine has further improved cancer treatment and developed many potential applications against cancer. With the deeper exploration of tumor pathogenesis, pharmacology, nanomedicine, etc., more intelligent and effective conjugated nanomedicines would be developed and transferred from the laboratory to the clinical application, finally benefiting more cancer patients and achieving victory over cancer in the near future.

## Figures and Tables

**Figure 1 pharmaceutics-14-01522-f001:**
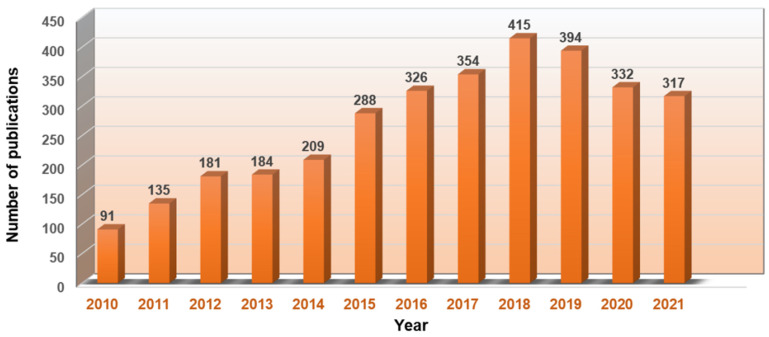
Number of publications per year on “conjugate nanomedicine” from 2010 to 2021, based on web of science database.

**Figure 2 pharmaceutics-14-01522-f002:**
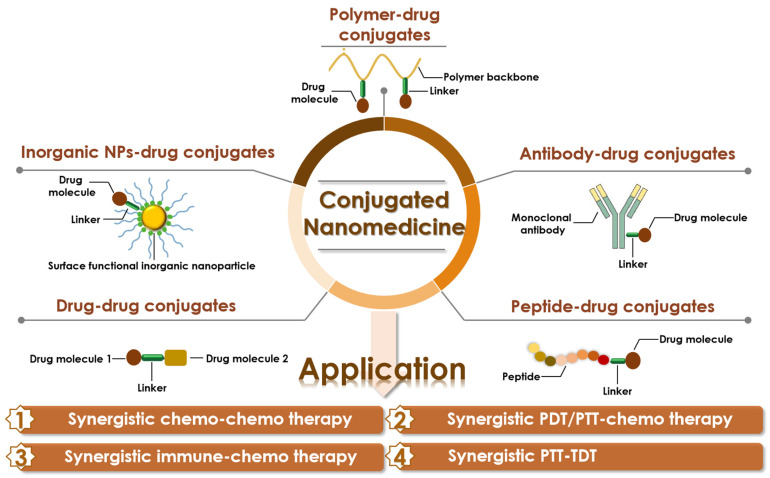
Overview of conjugated nanomaterials and representative clinical application of conjugated nanomaterials. (Photodynamic therapy, PDT; Photothermal therapy, PTT; Thermodynamic therapy, TDT).

**Figure 3 pharmaceutics-14-01522-f003:**
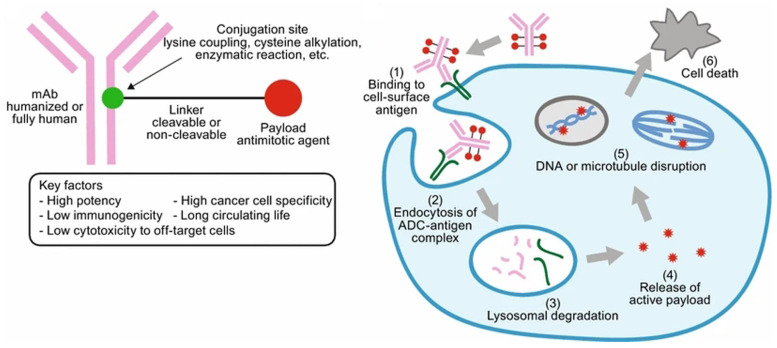
Structure and mechanism of action of ADC. Reprinted with permission from Ref. [49]. 2018, Springer.

**Figure 4 pharmaceutics-14-01522-f004:**
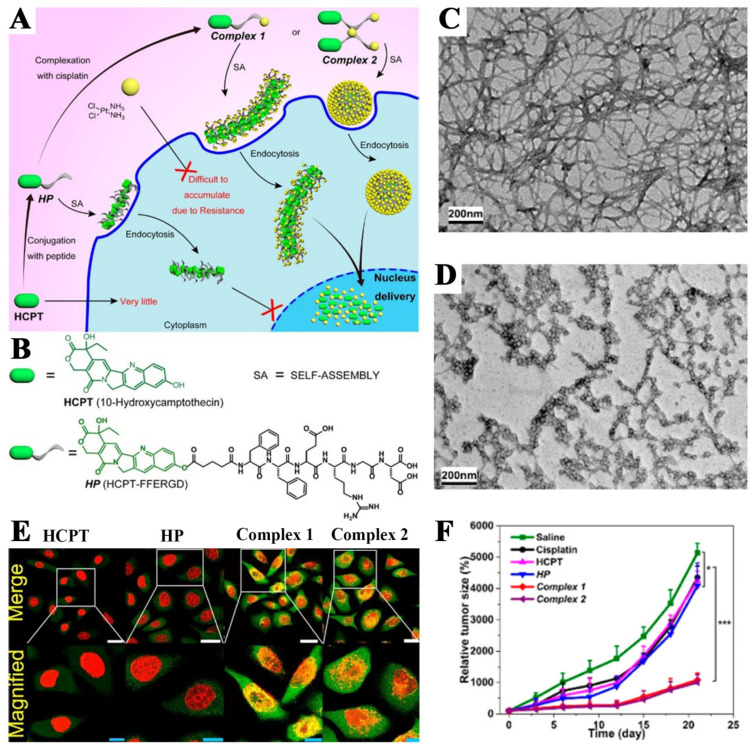
(**A**) Schematic illustration for preparation of dual-drug assemblies and the nuclear drug delivery, (**B**) Chemical structures of HCPT and HP, TEM image of solution containing 100 μM of (**C**) Complex 1, and (**D**) Complex 2, (**E**) CLSM images of A549 cells treated with HCPT, HP, Complex 1, and Complex 2 (100 μM) for 2 h, and then stained with 1 × Red dot 1. (**F**) in vivo anticancer efficacy, * *p* < 0.05 and *** *p* < 0.001. Reprinted with permission from Ref. [93]. 2017, American Chemical Society.

**Figure 5 pharmaceutics-14-01522-f005:**
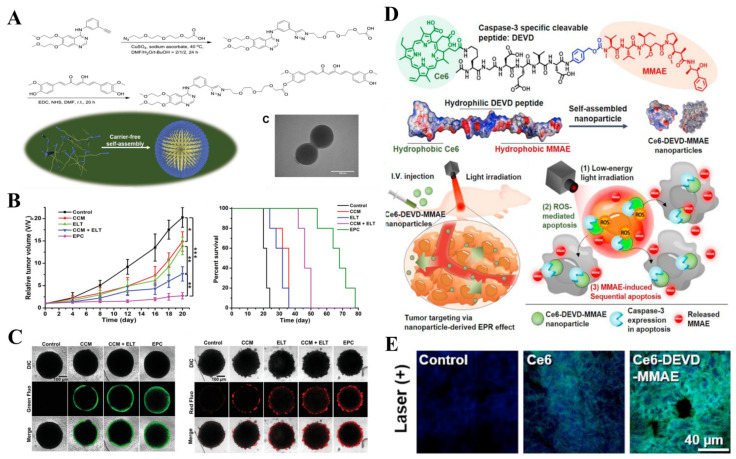
(**A**)The development of erlotinib–PEG–curcumin(EPC) nano-assembly and its characterization. (**B**) in vivo study of the EPC nano-assembly in a pancreatic xenograft mouse tumor model: quantitative fluorescence intensity of CCM and EPC in different organs and tumor growth profiles. * *p* < 0.05, ** *p* < 0.01 and *** *p* < 0.001. (**C**) Fluorescence images of drug penetration and cell killing effect in BxPC-3 tumor spheroids. Reprinted with permission from Ref. [99]. 2020, Wiley Online Library. (**D**) Schematic representation of visible light-induced apoptosis activatable nanoparticles of Ce6–DEVD–MMAE for targeted cancer therapy. (**E**) In vivo therapeutic efficacy of Ce6–DEVD–MMAE nanoparticles in tumor-bearing mice: Ex vivo apoptosis fluorescence imaging with Annexin V–Cy 5 (green color; Annexin V–Cy 5, blue color; DAPI). Reprinted with permission from Ref. [126]. 2019, Elsevier.

**Figure 6 pharmaceutics-14-01522-f006:**
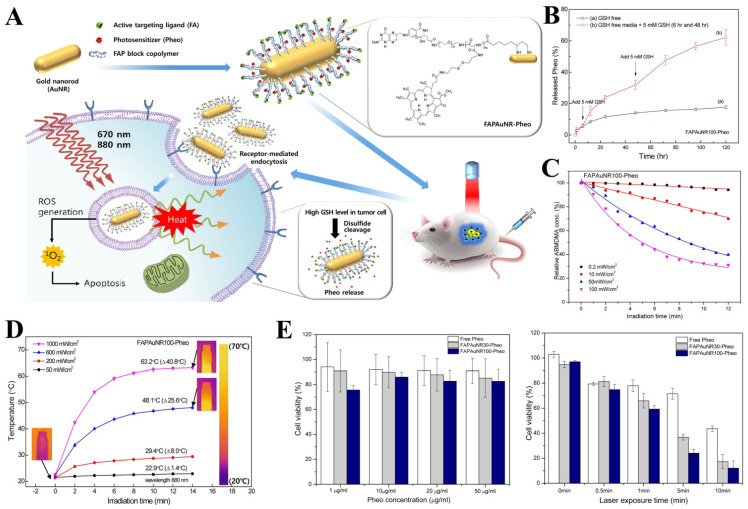
(**A**) Schematic representation of demonstrating the photothermally enhanced photodynamic therapy of GSH-responsive Pheo-conjugated AuNR. GSH-mediated Pheo release kinetics: (**B**) in vitro Pheo release profile of FAPAuNR100–Pheo with different GSH concentrations, (**C**) singlet oxygen generation behavior of free FAPAuNR100–Pheo at intensities of 0.2, 10, 50, and 100 mW/cm^2^, (**D**) photothermal effect of FAPAuNR100–Pheo at different 880 nm laser intensities, (**E**) Assay on the phototoxicity of free Pheo, FAPAuNR100–Pheo and FAPAuNR100–Pheo. Reprinted with permission from Ref. [137]. 2020, Elsevier.

**Figure 7 pharmaceutics-14-01522-f007:**
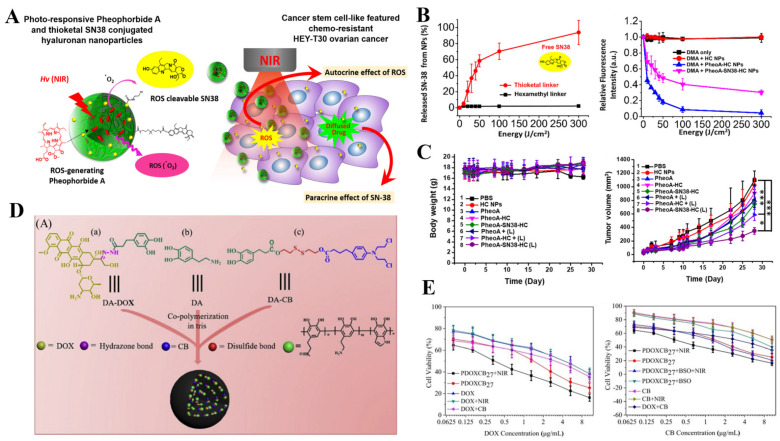
(**A**) Schematic representation of demonstrating combination of therapeutic hyaluronan nanoparticles conjugated with photodynamic pheophorbide A and ROS-cleavable thioketal-SN38 and drug delivery mechanism of nanoparticles. (**B**) Characterization of PhoeA-SN38-HC NPs: NIR induced singlet oxygen generation from NPs. In the presence of DMA (100 μM), 1 mg/mL NPs were exposed to light, and fluorescence intensity (λ = 420 nm) of DMA was measured by spectrometer, Light induced drug release from NPs depending on light energy. (**C**) In vivo PDT treatment with HC-PheoA-SN38: body weights and tumor growth curves of HEY-T30 xenograft BALB/C nude mouse, * *p* < 0.05 and *** *p* < 0.001. Reprinted with permission from [164]. Copyright@ Elsevier. (**D**) The preparation of dual drugs-conjugated PDOXCBs nanoparticles. (**E**) Both PDOXCB18 and PDOXCB27 cytotoxicity against cancer cells. Reprinted with permission from Ref. [167]. 2021, Elsevier.

**Figure 8 pharmaceutics-14-01522-f008:**
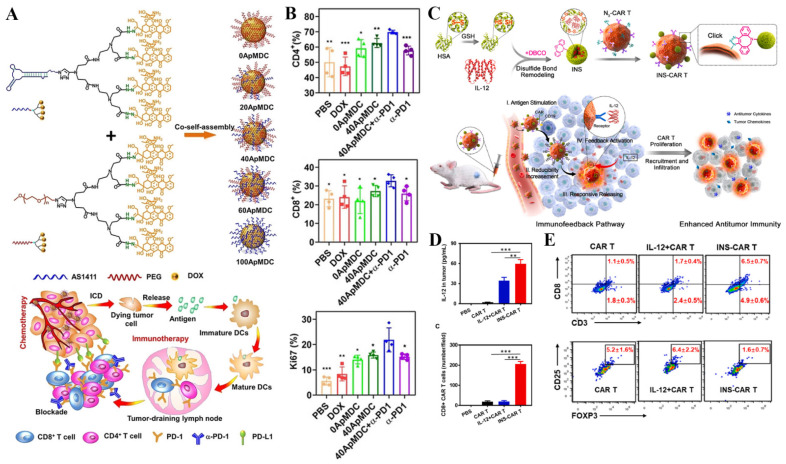
(**A**) Preparation of ApMDC and their self-assembled nanomicelles with tunable surface density of aptamers, initiation of antitumor immune responses of checkpoint blockade therapy by tumor-targeting, yet enhanced, chemotherapy. (**B**) In vivo antitumor immune responses: quantitative analysis of tumor infiltrating CD4+, CD8+, and level of Ki67 in the tumor-draining lymphoid node after treatment with PBS, free DOX, 0ApMDC, 40ApMDC, 40ApMDC + a-PD1 and a-PD1, respectively, * *p* < 0.05, ** *p* < 0.01 and *** *p* < 0.001. Reprinted with permission from Ref. [171]. 2021, Wiley-VCH GmbH. (**C**) Schematic illustration of IL-12 nanostimulant-engineered CAR T cells (INS-CAR T) biohybrids with immunofeedback to enhance immunotherapy in solid tumors. The CAR T cell-mediated INS delivery system elicited CAR T cell infiltration and enhanced immune responses. (**D**) IL-12 accumulation in tumor of NOD/SCID mice at 48 h post last administration and the average CAR T cell number in each field (600 μm × 600 μm) of tumor tissues was calculated from 50 fields, ** *p* < 0.01 and *** *p* < 0.001. (**E**) Representative flow cytometry analysis of the ratios of the percentages of CD8+ CAR T cells and coexpression of CD25 and Foxp3 among CD4+ CAR T cells in tumor tissues. Reprinted with permission from Ref. [176]. 2022, Elsevier.

**Table 1 pharmaceutics-14-01522-t001:** Nanomedicines for cancer treatment with granted regulatory approval [10,11,12,13,14,15].

Trade Name	Active Principle	Nanotechnology Platform	Indication	Approved Status
Doxil/Caelyx	Doxorubicin	PEGylated liposomes	Breast cancer, ovarian cancer, myeloma	FDA ^1^, EMA ^2^
DaunoXome	Daunorubicin	Liposomes	Kaposi sarcoma	FDA
Myocet	Doxorubicin	Liposomes	Breast cancer	FDA
Lipusu	Paclitaxel	Liposomes	breast cancer, non-small-cell lung cancer	NMPA ^3^
Abraxane	Paclitaxel	Albumin-bound nanoparticles	Metastatic breast cancer, metastatic pancreatic cancer, advanced non-small-cell lung cancer	FDA, EMA
Genexol-PM	Paclitaxel	Polymeric micelles	Non-small-cell lung cancer	KFDA ^4^
MEPACT	Mifamurtide	Liposomes	Osteosarcoma	EMA
Marqibo	Vincristine	Liposomes	Philadelphia chromosome-negative acute lymphoblastic leukemia	FDA
PICN	Paclitaxel	Polymer/lipid nanoparticles	Metastatic breast cancer	India
Onivyde (MM-398)	Irinotecan	PEGylated liposomes	Metastatic pancreatic cancer	FDA
VYXEOS	Cytarabine/daunorubicin (5:1)	Liposomes	Acute myeloid leukaemia	FDA, EMA
Paclical	Paclitaxel	Polymeric micelles	Ovarian cancer	Russia
Hensify	N/A	Hafnium oxide NP	Locally advanced soft tissue sarcoma	EMA
DHP107	Paclitaxel	Lipid nanoparticles	Advanced gastric cancer	KFDA
NanoTherm	N/A	Iron oxide nanoparticles	Recurrent glioblastoma	EMA
Nanoxel	Paclitaxel	Polymeric micelles	Breast cancer, ovarian cancer	India
Depocyt	Cytarabine	Liposomes	Acute Nonlymphocytic Leukemia, Meningeal Leukemia, Lymphomatous Meningitis	FDA

^1^ FDA: Food and Drug Administration. ^2^ EMA: European Medicines Agency. 3 NMPA: National Medical Products Administration. ^4^ KFDA: Korean Food and Drug Administration.

**Table 2 pharmaceutics-14-01522-t002:** ADCs on the market.

Trade Name	ADC	Target Antigen	Indication	Approved Status
Mylotarg	Gemtuzumab ozogamicin	CD33	CD33 positive AML	FDA
Adcetris	Brentuximab vedotin	CD30	Hodgkin lymphoma and anaplastic large cell lymphoma	FDA
Kadcyla	Ado-trastuzumab emtansine	HER2	HER2-positive breast cancer	FDA
Besponsa	Inotuzumab ozogamicin	CD22	Relapsed or refractory B cell precursor acute lymphoblastic leukemia	FDA
Lumoxiti	Moxetumomab pasudotox	CD22	Relapsed or refractory B-cell precursor acute lymphoblastic leukemia	FDA
Polivy	Polatuzumab vedotin	CD79b	Relapsed or refractory diffuse large B cell lymphoma	FDA
Padcev	Enfortumab vedotin	Nectin-4	Advanced or metastatic urothelial	FDA
Enhertu	Trastuzumab deruxtecan	HER2	Relapsed or refractory diffuse large B cell lymphoma	FDA
Trodelvy	Sacituzumab govitecan	Trop-2	HER2-triple-negative breast cancer	FDA
Blenrep	Belantamab mafodotin	BCMA	Relapsed or refractory multiple myeloma	FDA
Akalux	Cetuximab IRDye700DX	EGFR	Head and neck tumors, esophageal tumors, lung tumors, colon cancers	PMSB ^1^
Zvnionta	Loncastuximab tesirine	CD19	Relapsed or refractory diffuse large B cell lymphoma	FDA
Aidixi	Disitamab vedotin	HER-2	HER-2 positive metastatic gastric cancer	NMPA
Tivdak	Tisotumab vedotin-tftv	TF	Relapsed or metastatic cervical cancer	FDA

^1^ PMSB: Pharmaceutical and Medical Safety Bureau (Japan).

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
