# Peer review of "Research Progress of Conjugated Nanomedicine for Cancer Treatment"

_pharmaceutics, 2022, doi:10.3390/pharmaceutics14071522_

Round 1

Reviewer 1 Report

Manuscript ID: pharmaceutics-1811378 

Type of manuscript: Review

Title:  Research Progress of Conjugated Nanomedicine for Cancer Treatment

Comments to Editor and authors

In the manuscript entitled “Research Progress of Conjugated Nanomedicine for Cancer Treatment” the authors describes the conjugated nanomedicine and their various recent applications in synergistic chemotherapy.

Major revision:

There are some critical points in the review that should be better treated.  In the paragraph 2.2 Antibody-drug conjugates does not treated the topic regarding the ADCs made with reducible linkers and neither the bystander effect due to the reducing tumoral microenvironment. This part should be amplified.

Please see the literature below

·         Gébleux R, Wulhfard S, Casi G, Neri D. Antibody Format and Drug Release Rate Determine the Therapeutic Activity of Noninternalizing Antibody-Drug Conjugates. Mol Cancer Ther. 2015 Nov;14(11):2606-12. doi: 10.1158/1535-7163.MCT-15-0480. Epub 2015 Aug 20. PMID: 26294742; PMCID: PMC5606287.

 Moreover, the toxins of various origins used as payloads in different ADCs are not treated or even mentioned. Please see the literature below

·         Giansanti F, Flavell DJ, Angelucci F, Fabbrini MS, Ippoliti R. Strategies to Improve the Clinical Utility of Saporin-Based Targeted Toxins. Toxins (Basel). 2018 Feb 13;10(2):82. doi: 10.3390/toxins10020082. PMID: 29438358; PMCID: PMC5848183.

Minor points:

Page 2, lanes 64-74: point 2 is missing

Figure 1 (page 3, Lanes 87-88) : please specify the abbreviations PDT/PTT and PTT/TDT in the caption

Page 7, Lane 260 please correct myel with myeloid

Page 15, lane 562 please correct cm2 with cm2

For these reasons, after the corrections/additions requested the manuscript can be accepted. 

Reviewer 2 Report

 Bin Zhao and co-workers presented a review on "Research Progress of Conjugated Nanomedicine for Cancer Treatment".  They mainly focused on combined targeted nanomedicine methodologies: chemotherapy, photothermal & photodynamic therapy, thermal dynamic therapy and immunotherapy with the purpose of achieving synergistic effect during cancer treatment.

As to my opinion this review almost comprehensively described timely targeted anticancer methods and provides to readers "big picture" in the field. However, the photodynamic section (2.4.3. Photosensitizer-drug conjugates) lacks discussion on antibody driven PDT which is an emerging technique in treating cancer. I recommend publication after minor corrections:

1.      Table 1: "DepoCyt Ado-trastuzumab emtansine". DepoCyt is not an ADC but cytarabine liposome injection

2.      In section 2.4.3. "Photosensitizer-drug conjugates" the short discussion on ADCs conjugated to NIR photosensitizers should be added. The following references can assist and therefore should be added to the bibliography:

a.      Ebaston Thankarajan et al., 'Antibody guided activatable NIR photosensitizing system for fluorescently monitored photodynamic therapy with reduced side effects'ournal of Controlled Release, 343, 2022, 506-517, https://doi.org/10.1016/j.jconrel.2022.02.008

b.      Pye Hayley et al., 'Antibody-Directed Phototherapy (ADP)' Antibodies 2013, 2, 270-305; doi:10.3390/antib2020270

Reviewer 3 Report

The manuscript entitled ‘’Research Progress of Conjugated Nanomedicine for Cancer Treatment'' contains some interesting findings, and it may ultimately be suitable for publication. A significant effort was made to prepare the review, which is quite interesting and unique. This article is impressive for the reviewer and audience of the nanotechnology community as well as material science. This review would show a significant impact on the tissue engineering, nanomedicine, and materials science community. The reviewer recommends this work be published in Pharmaceutics after a revision.

The reviewer has the following comments  

1.       An infographic of the number of publications reported each year on conjugate nanomedicine is recommended

2.       A summarized table of clinical trials or translations of nanomaterials is recommended.

3.       The submitted manuscript requires significant improvement before I can recommend publication. This is due to an unclear scope of the manuscript, a mystifying structure, and the need for language proofreading. In the following, I will try to make concrete suggestions on how to improve this article.

*Clarify the scope of this work:
The field of nanomedicine is especially based on polymers, is extremely vast, and would likely require a review of its own. If the authors really want to include polymer-based nanoconjugates in this review, the most important works by Narsimha Mamidi, EV Barrera, Alex Elias Zuniga, and others should be cited in section 2.1 polymer-drug conjugates.

4.       *Give clear definitions:
I was struggling with the broad use of the term “nanomedicine” for all kinds of material geometries by the authors. They were not always clearly distinguishing if they were discussing nanoparticles, hydrogels, nanofibers, small molecules, peptides, dendrimers, polymers, etc.  

5.       *Improve the structure of each chapter:

Most chapters are listing examples from the literature without providing any coherent connection between the different examples, which makes the manuscript very tiring to read. I think that each chapter could significantly benefit from a short text on what the challenges in the respective application are. This could include the types of cells that need to be supported for the respective application or an explanation of the different types of material characteristics (porosity, stiffness, bioactive signals, etc.) that would be needed to provide adequate materials for the particular challenge. In essence, the authors should provide a problem statement on the challenge that needs to be overcome at the beginning of each chapter.

Round 2

Reviewer 1 Report

The manuscript after the additions/corrections can be accepted in the present form

Reviewer 3 Report

The authors have clarified all my comments and the quality of the revised manuscript is ameliorated. The revised manuscript can be acceptable in its current form.